# Molecular Hybridization Strategy on the Design, Synthesis, and Structural Characterization of Ferrocene-*N*-acyl Hydrazones as Immunomodulatory Agents

**DOI:** 10.3390/molecules27238343

**Published:** 2022-11-30

**Authors:** Laís Peres Silva, Ivanilson Pimenta Santos, Dahara Keyse Carvalho Silva, Bruna Padilha Zurita Claro dos Reis, Cássio Santana Meira, Marcos Venícius Batista de Souza Castro, José Maurício dos Santos Filho, João Honorato de Araujo-Neto, Javier Alcides Ellena, Rafael Gomes da Silveira, Milena Botelho Pereira Soares

**Affiliations:** 1Department of Life Sciences, State University of Bahia (UNEB), Salvador 41150-000, BA, Brazil; 2Gonçalo Moniz Institute, Oswaldo Cruz Foundation (IGM-FIOCRUZ/BA), Salvador 40296-710, BA, Brazil; 3Institute for Innovation in Advanced Health Systems (CIMATEC ISI SAS—University Center SENAI/CIMATEC), Salvador 41650-010, BA, Brazil; 4Laboratory of Design and Synthesis Applied to Medicinal Chemistry-SintMed®, Center for Technology and Geosciences, Federal University of Pernambuco, Recife 50740-521, PE, Brazil; 5Multiuser Laboratory of Structural Crystallography, Institute of São Carlos, University of São Paulo, São Carlos 13566-590, SP, Brazil; 6Department of Chemistry, Federal Institute of Goiás, Campus Ceres, Ceres 76300-000, GO, Brazil

**Keywords:** immunomodulation, endotoxic shock, acute peritonitis, *N*-acyl hydrazones, ferrocene

## Abstract

Immunomodulatory agents are widely used for the treatment of immune-mediated diseases, but the range of side effects of the available drugs makes necessary the search for new immunomodulatory drugs. Here, we investigated the immunomodulatory activity of new ferrocenyl-*N*-acyl hydrazones derivatives (**SintMed**(**141**–**156**). The evaluated *N*-acyl hydrazones did not show cytotoxicity at the tested concentrations, presenting CC_50_ values greater than 50 µM. In addition, all ferrocenyl-*N*-acyl hydrazones modulated nitrite production in immortalized macrophages, showing inhibition values between 14.4% and 74.2%. By presenting a better activity profile, the ferrocenyl-*N*-acyl hydrazones **SintMed149** and **SintMed150** also had their cytotoxicity and anti-inflammatory effect evaluated in cultures of peritoneal macrophages. The molecules were not cytotoxic at any of the concentrations tested in peritoneal macrophages and were able to significantly reduce (*p* < 0.05) the production of nitrite, TNF-α, and IL-1β. Interestingly, both molecules significantly reduced the production of IL-2 and IFN-γ in cultured splenocytes activated with concanavalin A. Moreover, **SintMed150** did not show signs of acute toxicity in animals treated with 50 or 100 mg/kg. Finally, we observed that ferrocenyl-*N*-acyl hydrazone **SintMed150** at 100 mg/kg reduced the migration of neutrophils (44.6%) in an acute peritonitis model and increased animal survival by 20% in an LPS-induced endotoxic shock model. These findings suggest that such compounds have therapeutic potential to be used to treat diseases of inflammatory origin.

## 1. Introduction

Throughout life, the human organism is exposed to a series of agents that can break homeostasis, whether they are pathogenic or not [1]. Among the pathogens, bacteria, fungi, parasites, and viruses stand out, while among the non-pathogenic agents there are trauma, exposure to toxic compounds, radioactivity, and smoke. Exposure to these agents culminates in the emergence of inflammation as the body responds to such harmful stimuli [2]. Inflammation is one of the body’s protective and fundamental reactions. It occurs in order to eliminate the source of the noxious stimulus or tissue injury to which the organism is being subjected and involves a series of cellular and molecular processes that aim to reestablish body homeostasis [3].

Despite its remarkable protective function, the inflammatory process, in addition to being well orchestrated, needs to be well controlled and properly terminated to prevent it from contributing to the emergence of metabolic disorders that may lead to diabetes and cancer, for example [4]. It is reported that unresolved inflammation can lead to the emergence of a series of inflammatory diseases that affect a large part of the population, such as asthma, rheumatoid arthritis, and atherosclerosis [5].

Currently, inflammation is controlled with two classes of drugs widely used in clinical practice: non-steroidal anti-inflammatory agents (NSAIDs) and glucocorticoids [6,7]. Despite their known effectiveness in the treatment of inflammatory disorders, the indiscriminate use of these drugs by the population has been increasing and, along with it, the incidence of adverse effects also grows [6,7,8,9]. The frequent use of non-steroidal anti-inflammatory drugs causes a series of unwanted reactions in the body such as gastrointestinal complications, cardiovascular diseases, and kidney diseases, while glucocorticoids can cause cardiovascular disease and disorders such as Cushing’s syndrome [8,9].

In this scenario, it is necessary to develop effective drugs with fewer adverse effects for the management of inflammation, and *N*-acyl hydrazones (NAHs) appear as promising alternatives. As a privileged structure in medicinal chemistry [10], the NAH scaffold is often found as a structural part of strong candidates for the control of inflammatory diseases due to their already reported action on macrophages and lymphocyte cells and in experimental models of immune diseases [11,12,13]. It is a Schiff base resulting from the condensation of carbonylated substances with hydrazides and constitutes a key pharmacophore for the binding and consequent inhibition of cyclooxygenases, acting as NSAIDs [14,15]. In addition, the NAH fraction provides greater stability and a safer inhibition of COX, and it is believed that this occurs from the relative hydrogen acidity of the amine group or its ability to stabilize free radicals [16,17]. As an important concept for the design of potentially bioactive compounds, molecular hybridization is a useful strategy based on the combination of pharmacophoric moieties of diverse substances to lead to a new molecule with an improved biological response when compared to the starting structural models [18,19].

Targeting the development of molecular hybrids that incorporate a second fragment of importance for the desired biological activity and remain structurally simple, the ferrocene (Fc) core was selected for the present studies, since Fc-bearing compounds are recognized for their importance in medicinal chemistry [20]. Ferrocene derivatives have been found to play an important role in the discovery of new immunomodulatory molecules [21,22,23,24], especially due to the Fc mechanisms of action, which are usually multi-modal, and rarely accessible with most organic pharmacophores. Some of the well-known mechanisms of action of ferrocene comprise direct protein inhibition, photoactivation with consequent singlet oxygen generation and cellular damage, metalation of macromolecules, and, the most common, redox activation and reactive oxygen species (ROS) formation, leading to cellular oxidative stress [25,26,27]. Due to the iron presence in the structure and its role in the oxidative biochemical processes, oxidative stress is the most important mechanism of action observed in bioactive ferrocene derivatives, so its association with other pharmacophoric groups has been proven to be an excellent strategy for the design of new potential biological active molecules [28]. Therefore, a series of ferrocene-*N*-acyl hydrazone (Fc-NAH) hybrids has been designed according to the concept in Figure 1, which fulfills our main goal of obtaining simple molecules with an accessible and easy synthetic route as well as the potential for further molecular modifications and biological studies.

Based on this premise, our group carried out the synthesis of new Fc-NAH derivatives and, in the present study, we have investigated their structural characterization by means of spectroscopic and crystallographic techniques, as well as their in vitro immunomodulatory activity, and also tested the effectiveness of the most active molecule **SintMed150** in murine models of endotoxic shock and acute peritonitis.

## 2. Results

The synthetic route to prepare the planned compounds **SintMed**(**141**–**156**), depicted in Figure 1, was based on a method developed by our research group [29]. Commercially available aldehydes **1a**–**p** have undergone silver (I)-mediated oxidation under basic conditions [30], leading to the corresponding carboxylic acids **2a**–**p** as solid materials with good yields. Aryl carboxylic methyl esters **3a**–**p** were easily obtained by means of Fischer esterification, and thus converted into the corresponding hydrazides **4a**–**p** under reflux in the presence of hydrazine hydrate in excess. The key intermediate hydrazides **4a**–**p** and ferrocenecarboxaldehyde were reacted in the presence of cerium (III) chloride heptahydrate (CeCl_3_·7H_2_O) as a catalyst under mild conditions, according to a method developed in our laboratory [31], in order to afford the Fc-NAH series **SintMed**(**141**–**156**) with excellent isolated yields, stereoselectivity, and high purity of crude products. The Fc-NAH were characterized using spectroscopic techniques such as nuclear magnetic resonance of hydrogen (^1^H NMR) and carbon-13 (^13^C NMR) and infrared spectroscopy (IR), as well as elemental analysis.

In addition to the assessed physical and spectroscopic data, attempts to reinforce the formation of the thermodynamically more stable *E-*isomers by means of the CeCl_3_-catalyzed synthesis of the Fc-NAH have included the crystallographic structure acquisition of some molecules of this series, with success for compound **SintMed149**, one of the more active molecules, confirming the proposed molecular structures and their purity. The *E*-isomer structure is clearly confirmed as observed in Figure 2.

Initially, the cytotoxicity of the molecules was tested in the J774 macrophage cell line. All evaluated ferrocenyl-*N*-acyl hydrazones did not show cytotoxicity at the tested concentrations, exhibiting CC_50_ values greater than 50 µM (Table 1). Under the same conditions, gentian violet, used as a positive control, presented a CC_50_ value equal to 0.8 µM.

The anti-inflammatory effect of the molecules was initially evaluated at a concentration of 40 µM in cultures of macrophages stimulated with LPS + IFN-γ by analysis of nitric oxide production. As can be seen in Table 1, all Fc-NAH derivatives modulate nitrite production, showing inhibition values between 14.4% and 74.2%. The most active molecules were the ferrocenyl-*N*-acyl hydrazones **SintMed149** and **SintMed150**, which showed inhibition values of 71.7% and 74.2% respectively. Under the same conditions, dexamethasone (Dexa) showed inhibition of 64.9% (Table 1).

To further explore the anti-inflammatory potential of compounds **SintMed149** and **SintMed150,** a new set of experiments was performed using peritoneal macrophages. Initially, the cytotoxicity of the investigated compounds was evaluated in peritoneal macrophages in the presence of LPS + IFNγ. As revealed in Figure 3, the molecules were not cytotoxic at any of the concentrations tested, as well as dexamethasone at 40 µM.

Next, the anti-inflammatory effect of both compounds was better evaluated using a concentration-response curve in peritoneal macrophages. As expected, macrophage activation with LPS plus IFNγ increased the amount of nitrite production (Figure 4). Treatment with **SintMed149** and **SintMed150** inhibited, in a concentration-dependent manner, the production of nitrite (*p* < 0.05). Interestingly, the inhibitory effect of **SintMed149** and **SintMed150** was also observed in the production of the pro-inflammatory cytokines TNF and IL-1β (*p* < 0.05) (Figure 5). Under the same conditions, dexamethasone, at a concentration of 10 µM, also promoted the reduction of these cytokines and nitrite (Figure 4 and Figure 5).

To investigate the immunosuppressive activity of **SintMed149** and **SintMed150**, the levels of IL-2, IL-4, and IFN-γ were evaluated in the supernatant of cultures of splenocytes stimulated with concanavalin A. As shown in Figure 6, stimulation with concanavalin A induced a significant increase in the production levels of the cytokines IL-2, IL-4, and IFN-γ. Treatment using the hydrazones **SintMed149** and **SintMed150** promoted a significant and concentration-dependent reduction of IL-2 and **SintMed150** also performed the same feat in the production of IFN-γ. However, the molecules did not significantly reduce IL-4 levels. Under the same conditions, dexamethasone significantly reduced the levels of IL-2, IL-4, and IFN-γ (Figure 6).

After the in vitro results, we investigated the toxicity effect of a single dose of the compound **SintMed150** in BALB/c mice. Administration of 50 or 100 mg/kg of **SintMed150** did not cause mortality or the appearance of any sign of toxicity in animals (Table 2). In addition, no difference in body weight was observed in animals treated with **SintMed150** when compared to vehicle-treated mice (Table 3).

Then, we tested the compound **SintMed150** in a murine model of endotoxic shock induced by a lethal dose of LPS. As shown in Figure 7, in comparison with the vehicle group, the animals treated with the compound **SintMed150** had a longer survival time, despite, on the third day, all the animals treated with the dose of 50 mg/kg having already died. On the fourth day, only 20% of the group treated with the 100 mg/kg dose survived, this finding being statistically significant (*p* < 0.05). Under the same conditions, dexamethasone, at a dose of 25 mg/kg, promoted a more significant survival rate (83.3%).

Lastly, the anti-inflammatory effect of **SintMed150** was evaluated in a murine model of carrageenan-induced acute peritonitis. As can be seen in Figure 8, animals stimulated with carrageenan and treated with vehicle solution had a high number of neutrophils in the peritoneal lavage. Compared to the vehicle-treated group, pretreatment with **SintMed150** at 100 mg/kg caused a reduction in neutrophil migration of 44.6%. Under the same conditions, dexamethasone, at a dose of 25 mg/kg, induced a reduction of 55% in neutrophil migration (Figure 8).

## 3. Discussion

The derivatives **SintMed**(**141**–**156**) were characterized by usual spectroscopic techniques, namely IR, ^1^H NMR, and ^13^C NMR, with structural assignments assisted by DEPT, HSQC, and HMQC experiments, confirming the structures and stereochemical features for each compound. ^1^H NMR analysis of crude Fc-NAH derivatives has revealed that only the *E*-isomer was formed, in agreement with previously reported outcomes for other compounds synthesized using this method. In addition to the mild condition and low reaction time, cerium (III) chloride catalysis has proved to be highly stereoselective, leading exclusively to the *E*-isomers [31]. All compounds from this series exhibit a clear and easy pattern of signals in agreement with previously reported works, mainly based upon the analysis of the signals observed for the amidic (–CONH–) and iminic (–N=CH–) groups, which are singlets [29]. However, *N*-acyl hydrazones may exist as conformers due to the influence of some structural features of the substituents linked to the amidic carbonyl. It was ascertained that electron-withdrawing effects acting on an aromatic ring, *ortho*-substituents, or the linkage to non-aromatic substituents can destabilize the resonance effects involving the –CONH– portion, favoring the possible emergence of rotamers. Signal duplication for Fc-NAH derivatives due to rotamery has been observed for the hydrogen atoms of the –CONH– and –N=CH– moieties and agreed with reported results from the literature [32,33]. Additionally, the Fc signals can also split due to the rotamery. The signal duplication does not follow a standard behavior, so different patterns can appear for different compounds, which are completely described in the Appendix A. The crystal structure of compound **SintMed149** shows the *E*-configuration, confirming the stereochemical analysis based on NMR studies. It is also important to notice that this specific molecule is not planar throughout the aryl-*N*-acyl hydrazone backbone, corroborating the works of Lopes et al. [32] and da Silva et al. [33], which explain the nature of the rotamery observed for this compound (Appendix A). Any structural and/or electronic factors disrupting the resonance are at the origin of the rotamery observed for some members of the Fc-NAH series.

*N*-acyl hydrazones have been widely used in medicinal chemistry due to their ability to act on several molecular targets and ease of synthesis; therefore, the development of new NAH hybrids is an attractive strategy for drug design and discovery [34]. Several studies report the ability of NAH to modulate cells and inflammatory mediators of the immune response for in vitro and in vivo models of immune disorders [35,36]. The association with the ferrocene scaffold is a recognized strategy to enhance biological responses [37] and was expected to work well in our approach.

In this study, the immunomodulatory potential of 16 new Fc-NAH derivatives was investigated. The evaluated molecules presented non-cytotoxicity in the tested concentration, which reinforces the safety profile of the class, previously demonstrated in several cell lines, such as mouse splenocytes from BALB/c mice and J774 macrophages [11,13,38]. Meira et al. (2018) [13], using the same method described in this report, investigated the cytotoxicity of 24 *N*-acyl hydrazones and obtained CC_50_ values between 17.8 and >100 µM in the same cell line used in this report.

Knowing that nitric oxide (NO) is a key mediator of the inflammatory response, it is essential to verify the modulation of its production performed by the new compounds [39], as disclosed in Table 1. It was found that all tested Fc-NAH hybrids reduced the production of NO; however, some structural features seem to affect the activity, depending upon the aromatic moiety. **SintMed141** (phenyl) has exhibited 19.7% of NO inhibition production and was chosen as comparing parameter. Inspecting Table 1, only one compound has presented a poorer activity, namely derivative **SintMed146** (3,5-di-*t*-butyl-4-hydroxyphenyl) with an inhibition of 14.4%. The influence of the bulkiness of the *t*-butyl group on the biological response seems to be fundamental, especially in comparison to compounds **SintMed151** (3,4,5-trihydroxyphenyl) and **SintMed152** (3,4,5-trihymethoxyphenyl), both similarly substituted but more active with 25.2% and 55.5% of inhibition, respectively. The substitution at the *meta* position of the phenyl ring reduces unequivocally the NO production inhibition, as observed for compounds **SintMed153** (2-hydroxy-5-nitrophenyl, 43.7%) and **SintMed154** (3-chlorophenyl, 45.2%), both with activities lower than 50%. Heterocyclic aromatic rings such as in Fc-NAH hybrids **SintMed143** (2-furanyl) with 49.2% and **SintMed145** (2-quinolinyl) with 49.6% and the ring-fused compound **SintMed155** (1-naphthyl) with 35.6% of NO inhibition suggest that this kind of substitution is related to moderate-to-poor biological responses. However, the *para* substitution at the phenyl ring seems to exert a positive effect on the inhibition of the NO production, as can be ascertained by compounds **SintMed144** (4-cyanophenyl, 55.2%) and **SintMed147** (4-trifluoromethylphenyl, 59.9%). A set of four molecules has disclosed the most important outcomes for the whole series. Bearing *ortho* substituents, compounds **SintMed142** (2-chlorophenyl, 49.3%), **SintMed148** (2-tolyl, 56.6%), **SintMed149** (2-bromophenyl, 71.7%), and **SintMed150** (2-phenoxyphenyl, 74.2%) have brought to light the importance of this substitution pattern to the enhancement of the biological response and suggested that the lipophilic character of the *ortho*-substituent is directly responsible for the inhibition of the NO production. The remarkable results found for **SintMed149** and **SintMed150** are higher than the control drug dexamethasone and arouse the interest of further investigation of their immunomodulatory profiles.

These data corroborate with previous investigations, in which *N*-acyl hydrazone derivatives have been shown to be effective in reducing nitric oxide production in macrophage cultures stimulated with LPS plus IFN-γ, at concentrations ranging from 2.5 to 30 μM [11,13]. The previously studied compound **SintMed65**, derived from NAH and tested by Meira and collaborators (2018) [13], was able to decrease the levels of pro-inflammatory cytokines (TNF-α and IL-1β) produced by macrophages during the inflammatory process. TNF-α is considered the “master regulator” of inflammatory responses, it is mainly produced by macrophages, and orchestrates the production of other inflammatory mediators, as well as macrophages and lymphocytes for injured tissues [40], while IL-1β has potent pro-inflammatory activity and is crucial for the body’s defense against infections and injuries [41]. Here, it was observed that compounds **SintMed149** and **SintMed150** also promoted a significant reduction in TNF and IL-1β production.

Evidence shows that the hydrazone fraction of the compounds has a pharmacophoric character for the inhibition of cyclooxygenase (COX) and that non-steroidal anti-inflammatory drugs containing the NAH fraction are less ulcerogenic [16,42]. The hydrazone derivative LaSSBio-1386 demonstrated the ability to inhibit the phosphorylation of the iκB protein, promoting a negative regulation of NF-κB, a relevant inflammatory pathway, whose inhibition is related to the decrease in cytokine production and inflammatory response [11]. Other derivatives such as *N*-pyrazoloyl hydrazone of isatin and *N*-thiopheneacetyl hydrazone of isatin decreased the translocation of NF-κB into the nucleus and suppressed the MAPK pathway, evidenced by a decrease in p-38, JNK, and ERK protein production, which interferes with the production of pro-inflammatory mediators [43]. In addition, compound 3a inhibited the activation of the TLR4 signaling pathway in macrophages that induces the activation of NF-κB [44]. These findings encourage further investigations with NF-κB and MAPK signaling pathways to understand the mechanism of action of ferrocene-*N*-acyl hydrazone.

Then, it was found that **SintMed149** and **SintMed150** decreased IL-2 and IFN-γ levels and neither of the two molecules reduced IL-4 levels. IL-2 promotes the growth, differentiation, and maturation of lymphocytes and IFN-γ promotes apoptosis in infected cells and activates macrophages and NK cells [45,46]. Moraes et al. (2018) [36] obtained similar results when investigating the anti-inflammatory potential of indole-*N-*acyl hydrazone derivatives in murine splenocyte cultures.

The endotoxic shock model was previously used by Guimarães and colleagues (2018) [13] to verify the survival of mice in the face of a lethal dose of LPS. Under the same conditions used in this study, the compound in question (LaSSBio-1386), at 50 and 100 mg/kg, promoted an animal survival of 50 and 85%, respectively. Previously, the anti-inflammatory activity of NAH derivatives from the inhibition of cell migration in murine models of the subcutaneous air pocket and carrageenan-induced acute peritonitis was demonstrated [21,22]. The data presented here suggest that Fc-NAH molecules have the potential to modulate the immune response in inflammatory conditions.

## 4. Materials and Methods

### 4.1. Chemistry

All solvents and reactants were purchased from Sigma-Aldrich (Merck KGaA, Darmstadt, Germany), Fluka (Carvalhaes, Alvorada, RS, Brazil), Vetec (Vetec Química Fina, Duque de Caxias, RJ, Brazil), and Acros Chemicals (Thermo Fisher Scientific, Waltham, MA, USA), and were used without further purification. The reactions’ progresses were monitored by thin-layer chromatography (TLC), performed onto glass-backed plates of silica gel 60 F254 with gypsum from Merck (Merck KGaA, Darmstadt, Germany), and all compounds were detected by ultraviolet light (254 nm). Melting points were determined with a capillary apparatus Gehaka PF 1500 Farma (Gehaka, São Paulo, SP, Brazil) and are uncorrected. NMR spectra were recorded at 400 MHz for hydrogen and 100 MHz for carbon, using a Varian UNMRS 400 spectrometer (Varian, Palo Alto, CA, USA), or at 300 MHz for hydrogen and 75 MHz for carbon nuclei, using a Varian Unitplus 300 NMR (Varian, Palo Alto, CA, USA). Analyses were determined in DMSO-*d_6_* with chemical shift values (δ) in parts per million (ppm) and coupling constants (*J*) in Hertz (Hz) and measured at 25 °C. ^1^H and ^13^C assignments were assisted by 2D experiments, such as DEPT full edit, HMBC, and HSQC. The description of the results was based on the IUPAC numbering and name recommendations. IR spectra were recorded on a Tensor27 FTIR spectrometer from Bruker (Bruker, Billerica, MA, USA) or a Spectrum 400 FTIR-FTNIR spectrometer from Perkin Elmer (Perkin Elmer, Waltham, MA, USA) with the samples being analyzed as KBr pellets. Elemental analyses were performed in a Perkin Elmer 2400 Series L elemental analyzer (Perkin Elmer, Waltham, MA, USA). All spectra are available in the Appendix A section.

### 4.2. Preparation of Aryl Carboxylic Acids **2a**–**p** [30]

In a 100 mL round-bottom flask, 20 mmol of silver nitrate was suspended in 60 mL of potassium hydroxide aqueous solution (7%), which was stirred for 5 min before 10 mmol of the appropriate aldehydes **1a***–***p** was added into the suspension. The mixture was stirred at 60 °C for 1 h, then cooled to room temperature, and filtered. The filtrate was acidified with a hydrochloric acid solution (10%) until the formation of a precipitate, which was cooled in an ice bath, filtered out, and dried under vacuum. The crude products were characterized by comparing their melting points with literature values as well as by ^1^H NMR analysis. Yields have ranged from 70 to 90%, and all solid products were directly used in the next step without further purification.

### 4.3. Preparation of Aryl Carboxylate Methyl Esters **3a**–**p** [29]

In a 100 mL round-bottomed flask, 6 mmol of crude carboxylic acids **2a***–***p** was placed with 30 mL of methanol. Then, 2 mL of concentrated sulfuric acid was added dropwise under vigorous stirring, and the solution was allowed to reflux overnight. After cooling to room temperature, the methanol was removed at a rotary evaporator to afford an oil, which was dissolved in ethyl acetate (30 mL), and extracted with saturated aqueous sodium carbonate (3 × 30 mL), followed by saturated sodium chloride (1 × 30 mL). The organic layer was dried over anhydrous sodium sulfate, filtered off, prior to solvent removal, and drying under vacuum to afford crude esters as oils or solids, in yields ranging from 80 to 95%. ^1^H NMR data were found to agree with literature reports so that crude products could undergo the next reaction.

### 4.4. Preparation of Aryl Carbo Hydrazides **4a**–**p** [29]

In a 50 mL round-bottomed flask, 5 mmol of the corresponding methyl esters **3a***–***p** and 2 mL of ethanol were placed. Then, 2 mL of hydrazine hydrate (55%) was dropped under stirring, and the solution was allowed to reflux overnight. After cooling to room temperature, the reaction mixture was placed in an ice bath and filtered to afford crude products as solids in yields from 70 to 90%. Based on ^1^H NMR data and melting point measurements, the solid hydrazides could be reacted directly in the next step.

### 4.5. Preparation of Ferrocenyl-N-acyl Hydrazones **SintMed**(**141**–**156**)

To a stirred suspension of 1 mmol of appropriate hydrazide and 1 mmol of ferrocenecarboxaldehyde in 10 mL of methanol was added 10 mol-% cerium (III) chloride heptahydrate and the reaction mixture was stirred at 40 °C during 10–30 min. The reaction’s completion was monitored by TLC. Once concluded, the heating was put away, and 10 mL of water was added to the medium. After standing at the refrigerator, vacuum filtration was carried out, and the solid was washed with cold water/ethanol 1:1 followed by cold water. ^1^H NMR analysis of all crude products confirmed their purity and, in a few cases, residual methanol (δ 3.16 ppm) can be observed. Recrystallization from dioxane/water mixture afforded the pure products for biological purposes. Yields, melting points, and spectroscopic and elemental analysis data are listed below for each compound.

(*E*)-*N*′-(Ferrocenylmethylidene)benzohydrazide **SintMed141**: *R*_f_ 0.50 (AcOEt/Hexanes 1:1), red powder, 0.93 mmol, 93%, mp 178.8–180.5 °C (from dioxane/H_2_O 1:1); IR (KBr, ν_max_ cm^−1^): 3441, 3226 (CONH), 3063 (Ar CH), 1646 (C=O), 1607 (C=C), 1557 (C=N); ^1^H NMR (400 MHz; DMSO-*d*_6_, δH ppm): 11.5 (s, 1H, CONH), 8.29 (s, 1H, N=CH), 7.89 (d, 2H, ^3^*J* = 6.9 Hz, Ar H-2,6), 7.57 (d, 1H, ^3^*J* = 6.8 Hz, Ar H-4), 7.52 (d, 2H, ^3^*J* = 6.8 Hz, Ar H-3,5), 4.66 (s, 2H, N=CH-Cp H-2,5), 4.46 (s, 2H, N=CH-Cp H-3,4), 4.24 (s, 5H, Cp-H); ); ^13^C NMR (100 MHz; DMSO-*d*_6_, δH ppm): 162.3 (1C, C=O), 149.0 (1C, N=CH), 133.6 (1C, Ar C-1), 131.3 (1C, Ar C-4), 128.3 (2C, Ar C-3,5), 127.4 (2C, Ar C-2,6), 78.8 (1C, N=CH-Cp C-1), 70.1 (2C, N=CH-Cp C-3,4), 68.9 (5C, Cp), 67.5 (2C, N=CH-Cp C-2,5); Anal Calcd for C_18_H_16_FeN_2_O: C, 65.08; H, 4.86; N, 8.43; found: C, 65.01; H, 4.80; N, 8.51.

(*E*)-*N*′-(Ferrocenylmethylidene)-2-chlorobenzohydrazide **SintMed142**: *R*_f_ 0.56 (AcOEt/Hexanes 1:1), red powder, 0.91 mmol, 91%, mp 156.8–158.1 °C (from dioxane/H_2_O 1:1); IR (KBr, ν_max_ cm^−1^): 3181 (CONH), 2993 (aliphatic CH), 1641 (C=O), 1598 (C=C), 1547 (C=N); ^1^H NMR (400 MHz; DMSO-*d*_6_, δH ppm, ≈1.8:1 rotamers mixture): 11.65 (s, CONH minor), 11.57 (s, 1H, CONH), 8.12 (s, 1H, N=CH), 7.88 (s, N=CH), 7.55–7.41 (m, 4H, Ar, and minor), 4.66 (s, 2H, N=CH-Cp H-2,5), 4.46 (s, 2H, N=CH-Cp H-3,4), 4.34 (s, N=CH-Cp H-2,5 minor), 4.31 (s, N=CH-Cp H-3,4 minor), 4.23 (s, 5H, Cp-H), 4.15 (s, Cp-H minor); ^13^C NMR (100 MHz; DMSO-*d*_6_, δC ppm, ≈1.8:1 rotamers mixture): 167.8 (CONH minor), 161.7 (1C, CONH), 149.1 (1C, N=CH), 144.5 (N=CH minor), 136.1 (Ar C-1 minor), 135.3 (1C, Ar C-1), 131.1, 130.3, 130.2, 129.8, 129.6, 129.2, 128.7, 128.5, 127.1, 126.7 (5C, Ar, and minor), 78.8 (N=CH-Cp C-1 minor), 78.5 (1C, N=CH-Cp C-1), 70.2 (2C, N=CH-Cp C-3,4), 69.7 (N=CH-Cp C-3,4 minor), 68.9 (5C, Cp), 68.8 (5C, Cp minor), 67.6 (2C, N=CH-Cp C-2,5), 67.1 (2C, N=CH-Cp C-2,5 minor); Anal Calcd for C_18_H_15_FeN_2_ClO: C, 58.97; H, 4.12; N, 7.64; found: C, 58.86; H, 4.18; N, 7.54.

(*E*)-*N*′-(Ferrocenylmethylidene)furan-2-ylcarbohydrazide **SintMed143**: *R*_f_ 0.46 (AcOEt/Hexanes 1:1), dark-red powder, 0.91 mmol, 91%, mp 220.8–222.5 °C (from dioxane/H_2_O 1:1); IR (KBr, ν_max_ cm^−1^): 3215 (CONH), 2925 (aliphatic CH), 1651 (C=O), 1606 (C=C); ^1^H NMR (300 MHz; DMSO-*d*_6_, δH ppm): 11.5 (s, 1H, CONH), 8.28 (s, 1H, N=CH); 7.92 (s, 1H, Furyl H-5), 7.25 (s, 1H, Furyl H-3), 6.68 (s, 1H, Furyl H-4), 4.64 (s, 2H, N=CH-Cp H-2,5), 4.44 (s, 2H, N=CH-Cp H-3,4), 4.22 (s, 5H, Cp-H); ^13^C NMR (75 MHz; DMSO-*d*_6_, δC ppm): 153.6 (1C, CONH), 149.1 (1C, N=CH), 146.8 (1C, Furyl C-2), 145.4 (1C, Furyl C-5), 114.3 (1C, Furyl C-3), 111.9 (1C, Furyl C-4), 78.7 (1C, N=CH-Cp C-1), 70.1 (2C, N=CH-Cp C-3,4), 68.9 (5C, Cp), 67.5 (2C, N=CH-Cp C-2,5); Anal Calcd for C_16_H_14_FeN_2_O_2_: C, 59.66; H, 4.38; N, 8.70; found: C, 59.57; H, 4.45; N, 8.78.

(*E*)-*N*′-(Ferrocenylmethylidene)-4-cyanobenzohydrazide **SintMed144**: *R*_f_ 0.50 (AcOEt/Hexanes 1:1), red powder, 0.98 mmol, 98%, mp 236.0–237.9 °C (from dioxane/H_2_O 1:1); IR (KBr, ν_max_ cm^−1^): 3354, 3229 (CONH), 3090 (Ar CH), 2227 (C≡N), 1650 (C=O), 1611 (C=C), 1565 (C=N); ^1^H NMR (300 MHz; DMSO-*d*_6_, δH ppm): 11.8 (s, 1H, CONH), 8.31 (s, 1H, N=CH), 8.05 (br s, 2H, Ar), 8.02 (br s, 2H, Ar), 4.67 (s, 2H, N=CH-Cp H-2,5), 4.45 (s, 2H, N=CH-Cp H-3,4), 4.22 (s, 5H, Cp-H); ^13^C NMR (75 MHz; DMSO-*d*_6_, δC ppm): 160.9 (1C, CONH), 150.2 (1C, N=CH), 137.6 (1C, Ar C-1), 132.4 (2C, Ar), 128.3 (2C, Ar), 118.2 (1C, CN), 113.7 (1C, Ar C-4), 78.4 (1C, N=CH-Cp C-1), 70.3 (2C, N=CH-Cp C-3,4), 68.9 (5C, Cp), 67.6 (2C, N=CH-Cp C-2,5); Anal Calcd for C_19_H_15_FeN_3_O: C, 63.89; H, 4.23; N, 11.76; found: C, 63.99; H, 4.18; N, 11.71.

(*E*)-*N*′-(Ferrocenylmethylidene)quinolin-2-ylcarbohydrazide **SintMed145**: *R*_f_ 0.66 (AcOEt/Hexanes 1:1), red powder, 0.91 mmol, 91%, mp 227.1–228.7 °C (from dioxane/H_2_O 1:1); IR (KBr, ν_max_ cm^−1^): 3238 (CONH), 3050 (Ar CH), 1664 (C=O), 1600 (C=C), 1533 (C=N); ^1^H NMR (400 MHz; DMSO-*d*_6_, δH ppm): 11.9 (s, 1H, CONH), 8.60 (d, 1H, ^3^*J* = 8.4 Hz, Quinoline H-4), 8.52 (s, 1H, N=CH), 8.21 (d, 1H, ^3^*J* = 8.0 Hz, Quinoline H-3), 8.20 (d, 1H, ^3^*J* = 8.0 Hz, Quinoline H-8), 8.11 (d, 1H, ^3^*J* = 8.4 Hz, Quinoline H-5), 7.92 (t, 1H, ^3^*J* = 7.2 Hz, Quinoline H-7), 7.75 (t, 1H, ^3^*J* = 7.6 Hz, Quinoline H-6), 4.69 (s, 2H, N=CH-Cp H-2,5), 4.48 (s, 2H, N=CH-Cp H-3,4), 4.27 (s, 5H, Cp-H); ^13^C NMR (100 MHz; DMSO-*d*_6_, δC ppm): 159.7 (1C, C=O), 150.6 (1C, N=CH), 150.0 (1C, Quinoline C-2), 145.8 (1C, Quinoline C-8a), 137.9 (1C, Quinoline C-4), 130.5 (1C, Quinoline C-7), 129.1 (1C, Quinoline C-8), 128.8 (1C, Quinoline C-4a), 128.12 (1C, Quinoline C-5), 128.09 (1C, Quinoline C-6), 118.9 (1C, Quinoline C-3), 78.7 (1C, N=CH-Cp C-1), 70.2 (2C, N=CH-Cp C-3,4), 69.0 (5C, Cp), 67.6 (2C, N=CH-Cp C-2,5); Anal Calcd for C_21_H_17_FeN_3_O: C, 65.82; H, 4.47; N, 10.96; found: C, 65.75; H, 4.52; N, 11.03.

(*E*)-*N*′-(Ferrocenylmethylidene)-3,5-*tert*-butyl-4-hydroxybenzohydrazide **SintMed146**: *R*_f_ 0.76 (AcOEt/Hexanes 1:1), red powder, 0.97 mmol, 97%, mp 256.2–258.4 °C (dec, from dioxane/H_2_O 1:1); IR (KBr, ν_max_ cm^−1^): 3627, 3610 (br OH), 3507, 3223 (CONH), 3082 (Ar CH), 2912 (aliphatic CH), 1640 (C=O), 1607 (C=C), 1556 (C=N); ^1^H NMR (400 MHz; DMSO-*d*_6_, δH ppm): 11.3 (s, 1H, CONH), 8.29 (s, 1H, N=CH), 7.63 (s, 3H, OH, and Ar H-2,6), 4.64 (s, 2H, N=CH-Cp H-2,5), 4.43 (s, 2H, N=CH-Cp H-3,4), 4.22 (s, 5H, Cp-H), 1.42 (s, 18H, 2x*^t^*Bu); ^13^C NMR (100 MHz; DMSO-*d*_6_, δC ppm): 163.3 (1C, C=O), 156.9 (1C, Ar C-4), 148.2 (1C, N=CH), 138.3 (2C, Ar C-3,5), 124.8 (1C, Ar C-1), 124.2 (2C, Ar C-2,6), 79.2 (1C, N=CH-Cp C-1), 69.9 (2C, N=CH-Cp C-3,4), 68.9 (5C, Cp), 67.4 (2C, N=CH-Cp C-2,5), 34.5 (2C, ***C***^t^Bu), 30.1 (6C, ^t^Bu); Anal Calcd for C_26_H_32_FeN_2_O_2_: C, 67.83; H, 7.01; N, 6.08; found: C, 67.89; H, 6.94; N, 6.13.

(*E*)-*N*′-(Ferrocenylmethylidene)-4-(trifluoromethyl)benzohydrazide **SintMed147**: *R*_f_ 0.74 (AcOEt/Hexanes 1:1), red powder, 0.99 mmol, 99%, mp 220.9–222.5 °C (from dioxane/H_2_O 1:1); IR (KBr, ν_max_ cm^−1^): 3355, 3201 (CONH), 3083, 3053 (Ar CH), 1657 (C=O), 1608 (C=C), 1564 (C=N); ^1^H NMR (400 MHz; DMSO-*d*_6_, δH ppm): 11.7 (s, 1H, CONH), 8.32 (s, 1H, N=CH), 8.10 (d, 2H, ^3^*J* = 5.2 Hz, Ar H-2,6), 7.90 (d, 2H, ^3^*J* = 6.0 Hz, Ar H-3,5), 4.68 (s, 2H, N=CH-Cp H-2,5), 4.47 (s, 2H, N=CH-Cp H-3,4), 4.24 (s, 5H, Cp-H); ^13^C NMR (100 MHz; DMSO-*d*_6_, δC ppm): 161.1 (1C, C=O), 150.0 (1C, N=CH), 137.4 (1C, Ar C-1), 131.3 (q, 1C, ^2^*J_FC_* = 31 Hz, Ar C-4), 128.3 (2C, Ar C-2,6), 125.3 (2C, Ar C-3,5), 126.1 (q, 1C, ^1^*J_FC_* = 200 Hz, CF_3_), 78.5 (1C, N=CH-Cp C-1), 70.2 (2C, N=CH-Cp C-3,4), 68.9 (5C, Cp), 67.6 (2C, N=CH-Cp C-2,5); ^19^F NMR (376 MHz; DMSO-*d*_6_, δF ppm): -61.2 (s, 1F, CF_3_); Anal Calcd for C_19_H_15_FeN_2_F_3_O: C, 57.03; H, 3.78; N, 7.00; found: C, 56.99; H, 3.83; N, 7.03.

(*E*)-*N*′-(Ferrocenylmethylidene)-2-methylbenzohydrazide **SintMed148**: *R*_f_ 0.55 (AcOEt/Hexanes 1:1), red powder, 0.93 mmol, 93%, mp 207.9–209.8 °C (from dioxane/H_2_O 1:1); IR (KBr, ν_max_ cm^−1^): 3174 (CONH), 3056 (Ar CH), 2991 (aliphatic CH), 1643 (C=O), 1600 (C=C), 1559 (C=N); ^1^H NMR (400 MHz; DMSO-*d*_6_, δH ppm, ≈7:3 rotamers mixture):11.4 (s, 1H, CONH, and minor), 8.15 (s, 1H, N=CH), 7.89 (s, N=CH minor), 7.43–7.29 (m, 4H, Ar H-3,4,5,6, and minor), 4.64 (s, 2H, N=CH-Cp H-2,5), 4.44 (s, 2H, N=CH-Cp H-3,4), 4.36 (s, N=CH-Cp H-2,5 minor), 4.31 (s, N=CH-Cp H-3,4 minor), 4.23 (s, 5H, Cp-H), 4.17 (s, Cp-H minor), 2.38 (s, 3H, CH_3_), 2.30 (s, CH_3_ minor); ^13^C NMR (100 MHz; DMSO-*d*_6_, δC ppm, ≈7:3 rotamers mixture): 170.5 (CONH minor), 164.4 (1C, CONH), 148.4 (1C, N=CH), 144.0 (N=CH minor), 136.1, 135.6, 135.4, 134.4, 130.4, 129.6, 129.4, 128.7, 127.24, 127.17, 125.4, 124.9 (6C, Ar, and minor), 78.9 (N=CH-Cp C-1 minor), 78.8 (1C, N=CH-Cp C-1), 70.0 (2C, N=CH-Cp C-3,4), 69.6 (N=CH-Cp C-3,4 minor), 68.8 (5C, Cp), 68.7 (Cp minor), 67.4 (2C, N=CH-Cp C-2,5), 67.0 (N=CH-Cp C-2,5 minor), 19.23 (CH_3_ minor), 19.18 (1C, CH_3_); Anal Calcd for C_19_H_18_FeN_2_O: C, 65.92; H, 5.24; N, 8.09; found: C, 65.99; H, 5.20; N, 8.15.

(*E*)-*N*′-(Ferrocenylmethylidene)-2-bromobenzohydrazide **SintMed149**: *R*_f_ 0.70 (AcOEt/Hexanes 1:1), red powder, 0.87 mmol, 87%, mp 163.4–165.8 °C (dec, from dioxane/H_2_O 1:1); IR (KBr, ν_max_ cm^−1^): 3190 (CONH), 3055 (Ar CH), 1656 (C=O), 1600 (C=C), 1562 (C=N); ^1^H NMR (400 MHz; DMSO-*d*_6_, δH ppm, ≈2:1 rotamers mixture): 11.6 (s, CONH minor), 11.5 (s, 1H, CONH), 8.11 (s, 1H, N=CH), 7.87 (s, N=CH minor), 7.71 (d, 1H, ^3^*J* = 8.0 Hz, Ar H-6), 7.68 (d, ^3^*J* = 8.0 Hz, Ar H-6 minor), 7.54–7.37 (m, 3H, Ar H-3,4,5, and minor), 4.66 (s, 2H, N=CH-Cp H-2,5), 4.46 (s, 2H, ^3^*J* = 2.8 Hz, N=CH-Cp H-3,4), 4.34 (s, N=CH-Cp H-2,5 minor), 4.31 (s, N=CH-Cp H-3,4 minor), 4.24 (s, 5H, Cp-H), 4.16 (s, Cp-H minor); ^13^C NMR (100 MHz; DMSO-*d*_6_, δC ppm): 162.5 (1C, CONH), 149.0 (1C, N=CH), 144.4 (N=CH minor), 137.5, 132.7, 131.9, 131.2, 130.2, 129.2, 128.5, 127.6, 127.1 (5C, Ar C-1,3,4,5,6, and minor), 119.4 (1C, Ar C-2), 78.9 (N=CH-Cp C-1 minor), 78.6 (1C, N=CH-Cp C-1), 70.2 (2C, N=CH-Cp C-3,4), 69.7 (N=CH-Cp C-3,4 minor), 68.91 (5C, Cp), 68.86 (Cp minor), 67.6 (2C, N=CH-Cp C-2,5), 67.1 (N=CH-Cp C-2,5 minor); Anal Calcd for C_18_H_15_FeN_2_BrO: C, 52.59; H, 3.68; N, 6.81; found: C, 52.52; H, 3.71; N, 6.90.

(*E*)-*N*′-(Ferrocenylmethylidene)-2-phenoxybenzohydrazide **SintMed150**: *R*_f_ 0.65 (AcOEt/Hexanes 1:1), red powder, 0.83 mmol, 83%, mp 177.9–179.5 °C (dec, from dioxane/H_2_O 1:1); IR (KBr, ν_max_ cm^−1^): 3308 (CONH), 3060 (Ar CH), 1661 (C=O), 1602 (C=C), 1541 (C=N); ^1^H NMR (400 MHz; DMSO-*d*_6_, δH ppm, ≈7:3 rotamers mixture): 11.45 (s, CONH minor), 11.38 (s, 1H, CONH), 8.14 (s, 1H, N=CH), 7.84 (s, N=CH minor), 7.67–6.98 (m, 9H, Ar and OPh, and minor), 4.62 (s, 2H, N=CH-Cp H-2,5), 4.42 (s, 2H, N=CH-Cp H-3,4), 4.39 (s, N=CH-Cp H-2,5 minor), 4.32 (s, N=CH-Cp H-3,4 minor), 4.20 (s, 5H, Cp-H), 4.12 (s, Cp-H minor); ^13^C NMR (100 MHz; DMSO-*d*_6_, δC ppm, ≈7:3 rotamers mixture): 167.9 (CONH minor), 161.1 (1C, CONH), 156.5, 153.0 (2C, Ar), 148.4 (1C, N=CH), 144.2 (N=CH minor), 131.7, 130.4, 130.2, 129.8, 129.6, 128.8, 128.6, 127.5, 123.6, 123.4, 123.1, 122.9, 120.1, 119.1, 118.4, 118.3, 118.0 (10C, Ar, OPh, and minor), 78.9 (N=CH-Cp C-1 minor), 78.7 (1C, N=CH-Cp C-1), 70.1 (2C, N=CH-Cp C-3,4), 69.7 (N=CH-Cp C-3,4 minor), 68.8 (5C, Cp), 68.7 (Cp minor), 67.5 (2C, N=CH-Cp C-2,5), 67.1 (N=CH-Cp C-2,5 minor); Anal Calcd for C_24_H_20_FeN_2_O_2_: C, 67.94; H, 4.75; N, 6.60; found: C, 67.87; H, 4.82; N, 6.65.

*(E*)-*N*′-(Ferrocenylmethylidene)-3,4,5-trihydroxybenzohydrazide **SintMed151**: *R*_f_ 0.10 (AcOEt), red powder, 0.96 mmol, 96%, mp 305.3–306.0 °C (dec, from dioxane/H_2_O 1:1); IR (KBr, ν_max_ cm^−1^): 3552–3000 (br, OH), 3225 (CONH), 3084 (Ar CH), 1633 (C=O), 1608 (C=C), 1570 (C=N); ^1^H NMR (300 MHz; DMSO-*d*_6_, δH ppm): 11.2 (s, 1H, CONH), 9.13 (s, 2H, 3,5-OH), 8.77 (s, 1H, 4-OH), 8.23 (s, 1H, N=CH), 6.90 (s, 2H, Ar H-2,6), 4.62 (s, 2H, N=CH-Cp H-2,5), 4.42 (s, 2H, N=CH-Cp H-3,4), 4.21 (s, 5H, Cp-H); ^13^C NMR (75 MHz; DMSO-*d*_6_, δC ppm): 162.5 (1C, C=O), 147.8 (1C, N=CH), 145.4 (2C, Ar C-3,5), 136.6 (1C, Ar C-4), 123.6 (1C, Ar C-1), 106.9 (2C, Ar C-2,6), 79.2 (1C, N=CH-Cp C-1), 69.9 (2C, N=CH-Cp C-3,4), 68.9 (5C, Cp), 67.4 (2C, N=CH-Cp C-2,5); Anal Calcd for C_18_H_16_FeN_2_O_4_: C, 56.87; H, 4.24; N, 7.37; found: C, 56.82; H, 4.31; N, 7.32.

(*E*)-*N*′-(Ferrocenylmethylidene)-3,4,5-trimethoxybenzohydrazide **SintMed152**: *R*_f_ 0.18 (AcOEt/Hexanes 1:1), red powder, 0.90 mmol, 90%, mp 237.8–239.5 °C (dec, from dioxane/H_2_O 1:1); IR (KBr, ν_max_ cm^−1^): 3235 (CONH), 3083 (Ar CH), 2942 (aliphatic CH), 1646 (C=O), 1608 (C=C), 1582 (C=N); ^1^H NMR (400 MHz, DMSO-d_6_, δH ppm): 11.4 (s, 1H, CONH), 8.32 (s, 1H, N=CH), 7.23 (s, 2H, Ar H-2,6), 4.66 (s, 2H, N=CH-Cp H-2,5), 4.45 (s, 2H, N=CH-Cp H-3,4), 4.23 (s, 5H, Cp-H), 3.86 (s, 6H, 3,5-OCH_3_), 3.73 (s, 3H, 4-OCH_3_); ^13^C NMR (100 MHz, DMSO-d_6_, δC ppm): 161.8 (1C, C=O), 152.6 (2C, Ar C-3,5), 149.1 (1C, N=CH), 140.2 (1C, Ar C-4), 128.7 (1C, Ar C-1), 105.0 (2C, Ar C-2,6), 78.8 (1C, N=CH-Cp C-1), 70.1 (2C, N=CH-Cp C-3,4), 68.9 (5C, Cp), 67.5 (2C, N=CH-Cp C-2,5), 60.0 (1C, 4-OCH_3_), (2C, 3,5-OCH_3_); Anal Calcd for C_21_H_22_FeN_2_O_4_: C, 59.73; H, 5.25; N, 6.63; found: C, 59.65; H, 5.22; N, 6.70.

(*E*)-*N*′-(Ferrocenylmethylidene)-3,4,5-trimethoxybenzohydrazide **SintMed153**: *R*_f_ 0.18 (AcOEt/Hexanes 1:1), yellow powder, yield 86%, mp 221.4–222.6 °C (dec, from dioxane/H_2_O 1:1); IR (KBr, ν_max_ cm^−1^): 3379, 3328, 3255 (CONH, OH), 3095 (Ar CH), 1630 (C=O), 1558 (C=N); ^1^H NMR (400 MHz; DMSO-*d*_6_, δH ppm, ≈2:1 rotamers mixture): 13.6 (s, 1H, CONH), 9.23 (s, 1H, Ar H-6), 9.11 (s, Ar H-6 minor), 8.37 and 8.35 (2s, 1H, Ar H-4 major and minor), 8.12 (s, 1H, N=CH), 7.34 (s, 1H, Ar H-3), 4.50 (s, 2H, N=CH-Cp H-2,5), 4.38 (s, 2H, N=CH-Cp H-3,4), 4.16 (s, 5H, Cp-H); ^13^C NMR (100 MHz; DMSO-*d*_6_, δC ppm): 167.1 (1C, CONH), 149.0 (1C, N=CH), 136.1 (1C, Ar C-2), 134.4 (1C, Ar C-5), 128.1 (1C, Ar C-4), 127.4 (1C, Ar C-6), 126.3 (1C, Ar C-1), 120.9 (1C, Ar C-3), 78.7 (1C, N=CH-Cp C-1), 69.9 (2C, N=CH-Cp C-3,4), 68.8 (5C, Cp), 67.5 (2C, N=CH-Cp C-2,5); Anal Calcd for C_18_H_15_FeN_3_O_4_: C, 54.99; H, 3.85; N, 10.69; found: C, 54.92; H, 3.84; N, 10.74.

(*E*)-*N*′-(Ferrocenylmethylidene)-3-chlorobenzohydrazide **SintMed154**: *R*_f_ 0.53 (AcOEt/Hexanes 3:7), red powder, yield 98%, mp 225.7–228.2 °C (dec, from dioxane/H_2_O 1:1); IR (KBr, ν_max_ cm^−1^): 3196 (CONH), 3063 (Ar CH), 1643 (C=O), 1608 (C=C), 1568 (C=N); ^1^H NMR (400 MHz, DMSO-d_6_, δH ppm): 11.6 (s, 1H, CONH), 8.29 (s, 1H, N=CH), 7.95 (s, 1H, Ar H-2), 7.86 (d, 1H, ^3^*J* = 6.8 Hz, Ar H-6), 7.65 (d, 1H, ^3^*J* = 7.6 Hz, Ar H-4), 7.55 (t, 1H, ^3^*J* = 7.6 Hz, Ar H-5), 4.67 (s, 2H, N=CH-Cp H-2,5), 4.46 (s, 2H, N=CH-Cp H-3,4), 4.23 (s, 5H, Cp); ^13^C NMR (100 MHz, DMSO-*d*_6_, δC ppm): 160.8 (C=O), 149.7 (N=CH), 135.6 (1C, Ar C-3), 133.1 (1C, Ar C-1), 131.2 (1C, Ar C-4), 130.3 (1C, Ar C-5), 127.1 (1C, Ar C-2), 126.2 (1C, Ar C-6), 78.6 (1C, N=CH-Cp C-1), 70.2 (2C, N=CH-Cp C-3,4), 68.9 (5C, Cp), 67.5 (2C, N=CH-Cp C-2,5); Anal Calcd for C_18_H_15_FeN_2_ClO: C, 58.97; H, 4.12; N, 7.64; found: C, 59.04; H, 4.09; N, 7.58.

(*E*)-*N*′-(Ferrocenylmethylidene)-1-naphtylcarbohydrazide **SintMed155**: *R*_f_ 0.68 (AcOEt/Hexanes 1:1), red powder, yield 93%, mp 233.0–234.4 °C (dec, from dioxane/H_2_O 1:1); IR (KBr, ν_max_ cm^−1^): 3185 (CONH), 3035 (Ar CH), 2977, 2845 (aliphatic CH), 1636 (C=O), 1617 (C=C), 1564 (C=N); ^1^H NMR (400 MHz; DMSO-*d*_6_, δH ppm): 11.7 (s, 1H, CONH), 8.24–8.22 (m, 1H, Naphtyl), 8.20 (s, 1H, N=CH), 8.08 (d, 1H, ^3^*J* = 8.0 Hz, Naphtyl), 8.03–8.00 (m, 1H, Naphtyl), 7.74 (d, 1H, ^3^*J* = 7.2 Hz, Naphtyl), 7.61–7.58 (m, 3H, Naphtyl), 4.68 (s, 2H, N=CH-Cp H-2,5), 4.47 (s, 2H, N=CH-Cp H-3,4), 4.26 (s, 5H, Cp-H); ^13^C NMR (100 MHz; DMSO-*d*_6_, δC ppm): 163.9 (C=O), 148.9 (N=CH), 133.1, 133.0, 130.2, 130.0, 128.2, 126.8, 126.3, 125.6, 125.1, 124.9 (10C, Naphtyl), 78.8 (1C, N=CH-Cp C-1), 70.1 (2C, N=CH-Cp C-3,4), 68.9 (5C, Cp), 67.5 (2C, N=CH-Cp C-2,5); Anal Calcd for C_22_H_18_FeN_2_O: C, 69.13; H, 4.75; N, 7.33; found: C, 69.21; H, 4.73; N, 7.36.

(*E*)-*N*′-(Ferrocenylmethylidene)-2,4-dimethoxybenzohydrazide **SintMed156**: *R*_f_ 0.32 (AcOEt/Hexanes 1:1), red powder, yield 87%, mp 123.9–126.6 °C (dec, from dioxane/H_2_O 1:1); IR (KBr, ν_max_ cm^−1^): 3223 (CONH), 3083 (Ar CH), 2836 (aliphtic CH), 1641 (C=O), 1603 (C=C), 1561 (C=N); ^1^H NMR (400 MHz; DMSO-*d*_6_, δH ppm): 11.4 (s, 1H, CONH), 8.30 (s, 1H, N=CH), 7.55 (d, 1H, ^3^*J* = 8.0 Hz, Ar H-5), 7.48 (s, 1H, Ar H-3), 7.07 (d, 1H, ^3^*J* = 8.8 Hz, Ar H-6), 4.65 (s, 2H, N=CH-Cp H-2,5), 4.44 (s, 2H, N=CH-Cp H-3,4), 4.23 (s, 5H, Cp-H), 3.84 (s, 3H, OCH_3_), 3.83 (s, 3H, OCH_3_); ^13^C NMR (100 MHz; DMSO-*d*_6_, δC ppm): 161.8 (C=O), 151.4 (1C, Ar C-4), 148.5 (1C, Ar C-2), 148.2 (N=CH), 125.6 (Ar C-1), 120.7 (1C, Ar C-5), 110.9 (1C, Ar C-6), 110.7 (1C, Ar C-3), 79.0 (1C, N=CH-Cp C-1), 70.0 (2C, N=CH-Cp C-3,4), 68.9 (5C, Cp), 67.4 (2C, N=CH-Cp C-2,5), 55.5 (2C, 2xOCH_3_); Anal Calcd for C_20_H_20_FeN_2_O_3_: C, 61.24; H, 5.14; N, 7.14; found: C, 61.16; H, 5.15; N, 7.09.

### 4.6. X-ray Crystallographic Analysis

Single crystals of compound **SintMed149** were grown by slow evaporation of an acetone/dichloromethane (1:1) solution. The X-ray diffraction experiment was performed at 100 K on a Rigaku Synergy-S diffractometer (Applied Rigaku Technologies, Inc., Austin, TX, USA) with Mo Kα radiation (λ = 0.71073 Å). The CrysAlisPro 1.171.42.67a program (CrysAlisPRO, Oxford Diffraction /Agilent Technologies UK Ltd., Yarnton, UK) was used for data collection, cell refinement, data reduction, and gaussian method absorption correction. The structure was solved and refined using the software SHELXT2015 [47] and refined by SHELXL2015 [48] hosted on the OLEX2 program [49]. All atoms, except hydrogen, were identified and refined by least-squares full matrix F^2^ with anisotropic thermal parameters. The crystallographic tables and the structure representations were generated by OLEX2. Details of the single-crystal X-ray diffraction experiment can be found in the Appendix A as well as on the CCDC database (www.ccdc.cam.ac.uk/, accessed on 12 September 2022) free of charge under deposit number 2214776.

### 4.7. Drugs

Dexamethasone (Sigma-Aldrich, St Louis, MO, USA), a synthetic glucocorticoid, was used as a positive control in immunomodulatory assays. Gentian violet (Synth, São Paulo, SP, Brazil) was used as a positive control in the cytotoxicity assays. All compounds were solubilized in dimethylsulfoxide (DMSO; PanReac, Barcelona, Spain) and diluted in Dulbecco’s modified Eagle medium (DMEM; Life Technologies, GIBCO-BRL; Waltham, MA, USA) for use in the in vitro assays. The final concentration of DMSO was less than 0.1% in all in vitro experiments. For in vivo assays, the compounds were solubilized in a solution containing 30% sorbitol (Sigma-Aldrich), 10% Tween 80 (Sigma-Aldrich), and 60% saline.

### 4.8. Animals

Male BALB/c mice, aged between 4–8 weeks, were provided by the animal breeding facility of Gonçalo Moniz Institute, Salvador, Brazil, and maintained in sterilized cages with controlled temperature (22 ± 2 °C) and humidity (55 ± 10%), water ad libitum, and receiving a balanced diet for rodent. The experiments and procedures with animals were approved by the institution’s committee on the ethical handling of laboratory animals (approved number: L-IGM-018/15).

### 4.9. Cytotoxicity to Mammalian Cells

J774 macrophages were seeded into 96-well plates at a cell density of 1 × 10^4^ cells/well in a DMEM medium supplemented with 10% fetal bovine serum (FBS; GIBCO) and 50 µg/mL of gentamicin (Life Technologies, Carlsbad, CA, USA) and incubated for 24 h at 37 °C and 5% CO_2_. After that time, each compound was added in six concentrations (50–1.56 µM), in triplicate, and incubated for 72 h. At the end of treatment, 20 µL/well of Alamar Blue (Invitrogen) was added to the plates for 4 h. using a SpectraMax 190 Microplate Reader (Molecular Devices, Sunnyvale, CA, USA).

As a positive control, gentian violet was used at concentrations ranging from 0.04 to 10 µM. CC_50_ values were calculated using data from three independent experiments.

The second set of experiments was performed using peritoneal macrophages activated with LPS (500 ng/mL, Sigma-Aldrich) and IFNγ (5 ng/mL; Sigma-Aldrich).

Peritoneal exudate macrophages were obtained by washing, with a cold DMEM medium, the peritoneal cavity of BALB/c mice 4–5 days after injection of 3% thioglycolate (Sigma-Aldrich) in saline (1.5 mL per mouse). Cells were seeded into 96-well plates at a cell density of 1 × 10^5^ cells/well in a DMEM medium supplemented with 10% FBS and 50 µg/mL of gentamicin and incubated for 24 h at 37 °C and 5% CO_2_. After that time, each sample was added (12.5, 25, and 50 µM), in triplicate and incubated for 72 h. At the end of treatment, 20 µL/well of Alamar Blue (Invitrogen) were added to the plates for 10 h. Colorimetric readings were performed at 570 and 600 nm using a SpectraMax 190 Microplate Reader (Molecular Devices, Sunnyvale, CA, USA).

### 4.10. Macrophage Cultures

Immortalized J774 macrophages (2 × 10^5^ cells/well) or peritoneal macrophages (2 × 10^5^ cells/well) were incubated in 96-well plates in a DMEM medium supplemented with 10% FBS and 50 µg/mL of gentamicin, in triplicate, stimulated or not with LPS (500 ng/mL) and IFN-γ (5 ng/mL), and treated or not with different concentrations of the evaluated ferrocenyl-*N*-acyl hydrazones. The cells were incubated for 4 or 24 h at 37 °C and 5% CO_2_. Cell-free supernatants were collected after 4 h (for TNFα measurement) and 24 h (for IL-1β, TNFα, and nitrite quantifications), and kept at −80 °C until use.

### 4.11. Splenocyte Cultures

BALB/c splenocyte suspensions were prepared in a DMEM medium supplemented with FBS and gentamicin. Splenocytes (5 × 10^6^ cells/well) were plated in 24-well plates, in triplicate, and stimulated or not with concanavalin A (Con A; 5 µg/mL, Sigma-Aldrich). To evaluate lymphocyte function, splenocytes were activated in the absence or presence of various concentrations of **SintMed149** and **SintMed150** (10, 20, and 40 µM). After 24 h of treatment, cell-free supernatants were collected and kept at −80 °C until use.

### 4.12. Assessment of Cytokine and Nitric Oxide Production

The dosage of cytokines IFN-γ, IL-1β, IL-2, IL-4, and TNFα was performed from the supernatant of peritoneal macrophages or splenocytes were determined by enzyme-linked immunosorbent assay (ELISA), using DuoSet kits from R&D Systems, according to the manufacturer’s instructions. Nitric oxide production was estimated in macrophage culture supernatants harvested at 24 h using the Griess method for nitrite quantification [50].

### 4.13. Acute Toxicity in Mice

Male BALB/c mice (6–8 weeks of age; *n* = 6/group) were randomized into three groups and treated orally with a single dose of **SintMed150** (50 or 100 mg/kg) or vehicle (solution containing 30% sorbitol, 10 % Tween 80, and 60% saline). Animals were monitored for signs of general toxicity for 2 weeks after treatment. Observations involved changes in eyes, fur, and skin, and the occurrence of tremors, salivation, convulsions, diarrhea, sleep, lethargy, and coma. Body weights were taken and recorded at days 0, 7, and 14.

### 4.14. LPS-Induced Endotoxin Shock

Groups of five male BALB/c mice (4 weeks of age) were used for the LPS lethality assays. Mice were treated with different doses of **SintMed150** (50 or 100 mg/kg), dexamethasone (25 mg/kg), or vehicle (solution containing 30% sorbitol, 10% Tween 80, and 60% saline), by intraperitoneal (i.p.) route. Ninety minutes later, animals were challenged with 600 µg of LPS (from serotype 0111:B4 *Escherichia coli*, Sigma-Aldrich) in saline, by i.p. route. The survival rate was then monitored daily during 4 days.

### 4.15. Induction of Acute Peritonitis in Mice

Male BALB/c mice (6–8 weeks of age; *n* = 6/group) were randomized into three groups and treated orally with **SintMed150** (100 mg/kg), dexamethasone (25 mg/kg), or vehicle (solution containing 30% sorbitol, 10 % Tween 80, and 60% saline) 24 and 1 h before the challenge. Next, animals were challenged with a 250 μL injection of carrageenan (1 mg/mL; intraperitoneal route). After 4 h, animals were euthanized and peritoneal exudates were harvested by peritoneal lavage using 2.5 mL of saline solution. Cells were centrifuged at 400× *g* for 10 min, at 4 °C. The pellet was resuspended in saline (1 mL). Total leukocytes in the peritoneal fluid were determined in a Neubauer chamber after dilution in Trypan blue stain. Differential counting of neutrophils was carried out in rapid panotype-stained cytospin preparations. A differential count of 300 cells was made in a blinded fashion and according to standard morphologic criteria.

### 4.16. Statistical Analysis

To determine the cytotoxicity concentration of 50% of J774 macrophages (CC_50_), we used non-linear regression. One-way analysis of variance and Newman–Keuls multiple comparison tests were employed by using Graph Pad Prism version 8.0.1 (Graph Pad Software; San Diego, CA, USA). Differences were considered significant when the values of *p* were < 0.05. The data are representative of at least two or three experiments.

## 5. Conclusions

The investigation of the immunomodulatory potential of the series Fc-NAH has led to the successful discovery of 16 new bioactive molecules designed on the basis of molecular hybridization, while some structure–activity relationships (SAR) have arisen from the analysis of the outcomes. A structural pattern seems to be closely associated with the most active compounds, namely the *ortho* substitution at the phenyl ring. Additionally, the presence of a phenyl ring appears to be essential for achieving better biological responses. The data presented here demonstrate that **SintMed** molecules can modulate the immune response in inflammatory conditions by decreasing the production of inflammatory mediators such as IFN-γ, IL-1β, IL-2, nitric oxide, and TNF-α, without affecting cell viability. Furthermore, oral administration of the **SintMed150** derivative significantly increased the survival rate of mice in a model of endotoxic shock induced by LPS and decreased inflammatory cell migration in a peritonitis model induced by carrageenan. These findings suggest that the class of ferrocene-*N*-acyl hydrazones has therapeutic potential and may be useful in the development of drugs to treat inflammatory conditions.

## Data Availability

Not applicable.

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
