# Peer review of "Molecular Hybridization Strategy on the Design, Synthesis, and Structural Characterization of Ferrocene-N-acyl Hydrazones as Immunomodulatory Agents"

_molecules, 2022, doi:10.3390/molecules27238343_

Round 1
Reviewer 1 Report
In the title of the manuscript is "molecular hybridisation". In that approach, both partners are equally important for biological activity. Yet, the authors do not provide an explanation why was ferrocene chosen as a partner to hydrazones.
Does oxidative stress play a role in the mechanism of action (could be possible due to ferrocene moiety)?
Synthesis of the compounds (scheme, conditions, explanations) should be included in the Results part of the manuscript (specially because there is "synthesis" in the title).
Author Response
Reviewer's remarks: In the title of the manuscript is "molecular hybridisation". In that approach, both partners are equally important for biological activity. Yet, the authors do not provide an explanation why was ferrocene chosen as a partner to hydrazones.
Author´s response: An appropriate paragraph and an additional Figure 1 dealing with this suggestion have been introduced and we believe that it is successful in explaining the role of ferrocene in our structural design.
Reviewer's remarks: Does oxidative stress play a role in the mechanism of action (could be possible due to ferrocene moiety)?
Author´s response: At this point Reviewer #1 comments about the importance of the ferrocenyl moiety for our work and claims that there is a need to establish a hypothesis dealing with the mechanism of action of the ferrocenyl-N-acyl hydrazones (Fc-NAH) investigated in this paper. In the manuscript’s introduction, we have stated the importance of the ferrocenyl portion for our strategy, as suggested, but no specific topic about the mechanism of action of the series Fc-NAH has been exploited in this work. Due to the pertinent observations of Reviewer #1, a piece of information on the different possible mechanisms of action responsible for the biological relevance of the ferrocenyl group has been inserted along with the corresponding references. However, no preliminary hypothesis about the way the Fc-NAH derivatives exert their activity can be stated at this moment. The focus of this work has been to investigate how molecular hybridization could lead to the discovery of new active immunomodulators and establish the structure-activity relationships between the most active molecules. This goal has been indeed achieved, opening the possibility of further studies to exploit new molecular improvements based on the findings of this work. We don’t feel comfortable speculating about the mechanism of action of our molecules at this moment but surely envisage deepening such questions in further research activities.
Reviewer's remarks: Synthesis of the compounds (scheme, conditions, explanations) should be included in the Results part of the manuscript (specially because there is "synthesis" in the title).
Author´s response: We thank Reviewer #1 for this suggestion. We have adjusted Scheme 1 and its description in the Results section, while detailed synthetic methodologies have been described in the Materials and Methods section.
Reviewer 2 Report
This paper reported the discovery of ferrocene-N-acyl hydrazones as immunomodulatory agents through molecular hybridization strategy. The potent compounds (SinMed149 and SinMed150) in the inhibition of nitrite production were further evaluated, showing inhibitory effects on the production of TNF-α, IL-1β, IL-2, and IFN-γ, but not IL-4. The animal study convinced the beneficial effects of the representative compound (SinMed150) in mice survival and the anti-inflammatory effect in a mice model of carrageenan-induced acute peritonitis. The topic is interesting and fits the scope of Molecules. The manuscript is well-organized, and the references are supportive. However, the key experiments are required to be done and the critical issues are required to be addressed before its publication on Molecules.
1. The reported N-acyl hydrazones (NAH), as immunomodulatory agents, should be provided with structures in a Figure in introduction section.
2. The use of ferrocenyl segment in the design of molecules is required to be well described.
3. The structure-activity relationships of ferrocenyl-N-acyl hydrazones in the inhibition of nitric oxide (NO) production are required to be discussed.
4. In Figure 1, the macrophage stimulation is suggested to be optimized, as there is no difference of cell viability between the conditions with and without LPS/IFN-γ.
5. The cells (simulated with LPS/IFN-γ) growth/viability curves (treated or untreated with the compounds) are required to be provided, to show the immunomodulatory effects of the representative compounds in cells.
6. In the animal study, the dosage of SintMed150 is as high as 100 mg/Kg. The toxicity of this compound in normal mice is required to be evaluated.
7. The molecular mechanisms (such as the targeting proteins, pathways, etc.) are suggested to be discussed.
8. In the discussion section, the second paragraph should be double-checked, which is not consistent to the results. In this paragraph, it says ‘SintMed149 and SintMed150 decreased IL-4 levels whereas only SintMed150 was able to significantly decrease IFN-γ levels and neither of the two molecules reduced IL-2 levels.’ However, in the results, SintMed149 and SintMed150 did not decrease IL-4 levels whereas only Dexa was able to significantly decrease IL-4 levels (Fig 4C, 4D).
Author Response
Reviewer's remarks: This paper reported the discovery of ferrocene-N-acyl hydrazones as immunomodulatory agents through molecular hybridization strategy. The potent compounds (SinMed149 and SinMed150) in the inhibition of nitrite production were further evaluated, showing inhibitory effects on the production of TNF-α, IL-1β, IL-2, and IFN-γ, but not IL-4. The animal study convinced the beneficial effects of the representative compound (SinMed150) in mice survival and the anti-inflammatory effect in a mice model of carrageenan-induced acute peritonitis. The topic is interesting and fits the scope of Molecules. The manuscript is well-organized, and the references are supportive. However, the key experiments are required to be done and the critical issues are required to be addressed before its publication on Molecules.
Authors’ response: We appreciate the reviewer for revising our manuscript and for the helpful comments. Below, we listed the answers to each question.
Reviewer's remarks: The reported N-acyl hydrazones (NAH), as immunomodulatory agents, should be provided with structures in a Figure in introduction section.
Authors’ response: Welcoming Reviewer #2’s suggestion, Figure 1 has been introduced and brings on the requested demand.
Reviewer's remarks: The use of ferrocenyl segment in the design of molecules is required to be well described.
Authors’ response: This demand was also made by Reviewer #1 and fulfilled in the Introduction section.
Reviewer's remarks: The structure-activity relationships of ferrocenyl-N-acyl hydrazones in the inhibition of nitric oxide (NO) production are required to be discussed.
Authors’ response: This was an especially important contribution coming from Reviewer #2 and we gave our best to offer a good interpretation of these data and establish a SAR analysis in agreement with our findings. Reviewer #2 can find the modifications in the Discussion section.
Reviewer's remarks: In Figure 1, the macrophage stimulation is suggested to be optimized, as there is no difference of cell viability between the conditions with and without LPS/IFN-γ. The cells (simulated with LPS/IFN-γ) growth/viability curves (treated or untreated with the compounds) are required to be provided, to show the immunomodulatory effects of the representative compounds in cells.
Authors’ response: This result is expected considering that the concentrations of LPS and IFNy used by us aim only to activate the macrophage to have an M1 phenotype without exerting a cytotoxic effect. The objective of the cytotoxicity assay treating macrophages with hydrazones in the presence of LPS + IFNy is to prove that there is no cytotoxicity using conditions similar to what happens in assays that aim to assess anti-inflammatory activity.
Reviewer's remarks: In the animal study, the dosage of SintMed150 is as high as 100 mg/Kg. The toxicity of this compound in normal mice is required to be evaluated.
Authors’ response: In order to meet the reviewer's request and improve the quality of our work, a new set of experiments was carried out to evaluate the acute toxicity of SintMed150 at doses of 100 and 50 mg/Kg. The new data are shown in tables 2 and 3 of the new version of the manuscript.
Reviewer's remarks: The molecular mechanisms (such as the targeting proteins, pathways, etc.) are suggested to be discussed.
Authors’ response: Based on the reviewer's comment, the manuscript's discussion has been improved.
Reviewer's remarks: In the discussion section, the second paragraph should be double-checked, which is not consistent to the results. In this paragraph, it says ‘SintMed149 and SintMed150 decreased IL-4 levels whereas only SintMed150 was able to significantly decrease IFN-γ levels and neither of the two molecules reduced IL-2 levels.’ However, in the results, SintMed149 and SintMed150 did not decrease IL-4 levels whereas only Dexa was able to significantly decrease IL-4 levels (Fig 4C, 4D).
Authors’ response: The reviewer is correct, we are sorry for the mistake. The paragraph in question has been revised.
Round 2
Reviewer 1 Report
The authors have improved the manuscript according to the suggestions and I have no more comments.
Reviewer 2 Report
After the authors’ revision according to my and others’ previous comments, the quality of this paper was significantly improved, and could reach the required quality standard for Molecules in my opinion. I suggest accepting it without further revision.